



# An Electrochemical cell for *in operando* [13]C NMR investigations of carbon dioxide/carbonate processes in aqueous solution

Sven Jovanovic[1,5], Philipp Schleker[1,3], Matthias Streun[2], Steffen Merz[1], Peter Jakes[1], Rüdiger-A. Eichel[1,4], and Josef Granwehr[1,5]

[1]Forschungszentrum Jülich, Institute of Energy and Climate Research, Fundamental Electrochemistry (IEK-9), Jülich, Germany
[2]Forschungszentrum Jülich, Central Institute of Engineering and Analytics, Electronic Systems (ZEA-2), Jülich, Germany
[3]Department of Heterogeneous Reactions, Max Planck Institute for Chemical Energy Conversion, Mülheim an der Ruhr, Germany
[4]RWTH Aachen University, Institute of Physical Chemistry, Aachen, Germany
[5]RWTH Aachen University, Institute of Technical and Macromolecular Chemistry, Aachen, Germany

**Correspondence:** Sven Jovanovic (s.jovanovic@fz-juelich.de)

**Abstract.** *In operando* NMR spectroscopy is a method for the online investigation of electrochemical systems and reactions. It allows a real-time observation of the formation of products and intermediates, and it grants insight into the interactions of substrate and catalyst. An *in operando* NMR setup for the investigation of the electrolytic reduction of $CO_2$ on silver electrodes has been developed. The electrolysis cell consists of a three-electrode setup using a working electrode of pristine silver, a chlorinated silver wire as reference electrode, and a graphite counter electrode. The setup can be adjusted for the use of different electrode materials and fits inside a 5 mm NMR tube, Additionally, a shielding setup was employed to minimize noise caused by interference of external radio frequency (RF) waves with the conductive components of the setup. The electrochemical performance of the *in operando* electrolysis setup was investigated in comparison to a standard $CO_2$ electrolysis cell. The small cell geometry impedes the release of gaseous products, and thus it is primarily suited for current densities below 1 mA/cm[2]. The effect of conductive components on [13]C NMR experiments was studied using a $CO_2$ saturated solution of aqueous bicarbonate electrolyte. Despite the $B_0$ field distortions caused by the electrodes, line widths of *ca.* 1 Hz could be achieved. This enables the investigation of interactions in the sub-Hertz range by NMR spectroscopy. It was found that the dynamics of the bicarbonate electrolyte change due to interaction with the electrochemical setup, by catalyzing the exchange reaction between $CO_2$ and $HCO_3^-$ and affecting the formation of an electrical double layer.

## 1 Introduction

The anthropologically driven atmospheric $CO_2$ increase is considered one of the major contributions to global warming (Rockstrom et al., 2009; Hansen et al., 2008). A decline in anthropological $CO_2$ emissions is seen as improbable due to socio-economic factors (Grundmann, 2016). Thus, the recycling of $CO_2$ by electrochemical conversion to energy-rich materials is of particular interest (Jhong et al., 2013; Olah et al., 2009; Whipple and Kenis, 2010; Zhu et al., 2016; Higgins et al., 2018). A promising candidate in terms of cost and variability is the electrolytic reduction of $CO_2$, which is usually performed in an



aqueous bicarbonate electrolyte (Haas et al., 2018). Depending on the metal electrode, $CO_2$ electrolysis yields a number of products, *e.g.* formate, hydrocarbons, alcohols and carbon monoxide (Hori, 2008; Hori et al., 1989, 1994; Burdyny and Smith, 2019; Garg and Basu, 2015; Haas et al., 2018). CO, which is versatile educt for the chemical industry, *e.g.* as feedstock in the Fischer-Tropsch process, is produced at silver or gold electrodes (Hernández et al., 2017).

Despite vivid research, the reaction pathways of electrochemical $CO_2$ reduction are still not well understood (Hori, 2008; Jhong et al., 2013; Kortlever et al., 2015). There are two main issues. First, there is a complex equilibrium of $CO_2$ and carbonate species in aqueous systems, which depend on pH, temperature and partial pressure. These parameters vary not only with time during the electrolysis but also spatially as a function of distance from the electrode surface (Hori, 2008; Wang et al., 2010; Xiang et al., 2012; Gibbons and Edsall, 1963; Holmes et al., 1998; Jiao and Rempe, 2011; Lucile et al., 2012; Mani et al., 2006;

Toan et al., 2019). Second, the electrolytic $CO_2$ reduction suffers further from a high overpotential, which can be mitigated by a few selected metal catalysts. There is evidence that the cause of the overpotential is the formation of a $CO_2^-$ radical as an intermediate. (Hori, 2008; Janik and Tripathi, 2016; Baruch et al., 2015).

    To gain inside into the dynamic processes of an electrochemical system it is imperative to monitor the system as a whole during operation, *i.e.* using *in operando* spectroscopic techniques (Bañares, 2005; Britton, 2014). NMR spectroscopy is a

flexible and powerful method for chemical and reaction analysis (Williamson et al., 2019; Stanisavljev et al., 1998). The NMR study of batteries are often associated with broad line widths inherent to solid state materials. For *in operando* investigations of liquid state electrolysis systems, high spectral resolution is a limiting factor. The determination of structural information for small molecules relies on the visibility of minor changes in chemical shift as well as J-couplings in the range of a few Hertz, which are not visible in case of broad signals. Thus, signal line width was of concern since the earliest publication in this field.

Several experiment setups for the electrochemical reduction and/or oxidation of organic molecules are published (Bussy and Boujtita, 2015; Falck and Niessen, 2015). The first *in operando* flow cell for the investigation of electrochemical processes was described by Richards *et al.* in 1975 and consisted of a two-electrode setup inside a 5 mm NMR tube (Richards and Evans, 1975). A Pt/Hg wire working electrode outside the sensitive volume was contained in a 3 mm tube concentric to the NMR tube. At the bottom of the 3 mm tube, a capillary releases the reaction products into the sensitive volume of the 5 mm tube. The

setup allowed sample spinning, which was required due to the low spectral resolution of the spectrometer at that time. This approach was adapted by Mairanovsky *et al.* in 1983 for the investigation of anion radical decays and improved by using three electrodes (Mairanovsky et al., 1983).

    A new approach for an *in operando* setup was published in 1990 by Mincey et al. (Mincey et al., 1990). Thin film electrodes were employed to minimize distortions of the $B_0$ and $B_1$ field by the conductive parts of the electrolysis cell, and thus $^1$H line

widths of 0.9 Hz could be achieved (Prenzler et al., 2000; Webster, 2004; Zhang and Zwanziger, 2011). As a further evolution of the setup, Webster *et al.* introduced radio frequency (RF) chokes to minimize interactions between NMR and the potentiostat (Webster, 2004). However, due to its prerequisites manufacturing of thin film electrodes is not easily adaptable. An alternative setup was constructed by Klod *et al.*, which aimed for accessibility for non-specialized NMR users (Klod et al., 2009). The electrolysis cell employed carbon fiber electrodes with high surface area, and could be set up without the need for special



equipment (Bussy et al., 2013). However, the use of carbon fiber electrodes limits the variety in electrocatalysts that can be investigated.

  A different technique for the coupling of electrochemistry and magnetic resonance is hyphenated electrochemical NMR, first presented by Albert et al. in 1987 and later continued by other groups (Albert et al., 1987; Falck et al., 2013; Simon et al., 2012), where the electrochemical cell is physically separated from the NMR spectrometer by passing the electrochemically

generated species to an NMR probe by flow. This technique does not suffer from $B_0$ and $B_1$ distortions, but there is a time delay between generation and detection of the electrochemical species due to the physical separation.

  Most research in this field has been done using $^1$H NMR due to its high sensitivity compared to other nuclei, with only few attempts made to investigate $^{13}$C systems (Albert et al., 1987; Nunes et al., 2014). $^{13}$C NMR offers a high spectral width and thus increased separation between signals, but suffers from a low natural abundance of the nucleus. In a recent study,

Nunes *et al.* developed a setup based on Klod *et al.* to investigate the electrochemical reduction of 9-chlroantracene by $^{13}$C NMR (Nunes et al., 2014). They employed steady-state free precession (SSFP) to achieve high signal-to-noise ratio during short measurement times despite using non-enriched samples. Instead of NMR, electron paramagnetic resonance (EPR) spectroscopy was employed in a recent study by Neukermans *et al.* as a screening tool for electrocatalysts (Neukermans et al., 2020).

  Despite the appearance in 1975, *in operando* NMR investigations of liquid state systems are still a niche application with

focus on method development and thus tested on well-studied, simple redox systems (Richards and Evans, 1975; Mairanovsky et al., 1983; Albert et al., 1987; Mincey et al., 1990; Prenzler et al., 2000; Webster, 2004; Klod et al., 2009; Zhang and Zwanziger, 2011). Only in recent years *in operando* NMR has been used to study biological systems (Zhang and Zwanziger, 2011; Bussy et al., 2013; Falck et al., 2013). However, this method has not yet been utilized for the investigation of industrial and energy applications, *e.g.* the electrolytic reduction of $CO_2$.

*In operando* $^{13}$C NMR spectroscopy is ideally suited to study the electrolytic reduction of $CO_2$ to $CO$, which requires high resolution to monitor changes in educt structure and the ability to use high sensitivity NMR equipment. To directly measure processes of interest, the working electrode needs to be placed in the sensitive volume of the NMR coil. On the other hand, conductive components in the sample lead to distortions of $B_0$ and $B_1$. These effects can be minimized by choosing a proper placement and orientation of the electrode and by pulse sequences that are robust against $B_0$ and $B_1$ field distortions

(Romanenko et al., 2014; Hargreaves et al., 2011; Jungmann et al., 2017; Britton et al., 2013). For a versatile cell setup, ease of construction, adaptability to other metal electrodes and the applicability in unmodified NMR liquid state probes is desirable.

  This work aims to reduce the effort required for the construction of an *in operando* NMR setup and apply it to investigate the $CO_2$ electrolysis on a molecular level. Thus, an electrolysis cell for the *in operando* NMR investigation of electrolytic $CO_2$ reduction is presented. The cell is constructed inside a 5 mm NMR tube and consists of a three-electrode setup, which can

be adapted without the need of special tools. The setup was evaluated for electrochemical performance by characterizing the chemical system of $CO_2$ in 1M $KHCO_3$ electrolyte with and without electrochemical equipment connected. $T_1$ and $T_2$ as well as the exchange time between $CO_2$ and $HCO_3^-$ were determined to investigate the mobility and interactions of the reactant and electrolyte molecules.




## 2  *In operando* NMR setup

### 2.1  Electrolysis cell

A three-electrode electrolysis cell has been build that fits a standard 5 mm NMR tube. It consists of a 2.5 x 4 x 0.05 mm silver foil (GoodFellow, Hamburg, Germany) with an area of 10 mm$^2$ as working electrode, a graphite rod of 1 mm in diameter and 50 mm in length (GoodFellow, Hamburg, Germany) as counter electrode. A chlorinated silver wire tip with a diameter of 0.25 mm (GoodFellow, Hamburg, Germany) was employed as micro Ag/AgCl reference electrode. All electrodes were connected to a silver wire with a diameter 0.25 mm insulated using polytetrafluoroethylene (PTFE) (GoodFellow, Hamburg, Germany) with an insulation thickness of 0.024 mm. The graphite counter electrode was chosen because metals can dissolve in small quantities during electrolysis and deposit at the working electrode resulting in a change of catalytic properties (Benke and Gnot, 2002). This process is more pronounced in small setups with half cell reactions that are not separated by a membrane, as it is the case for the *in operando* electrolysis cell, since species from the counter electrode diffuse sufficiently fast towards the working electrode.

To join the silver lead wire and the silver foil used as working electrode, first the wire insulation was stripped off over a length of about 1–2 mm. Afterwards, the uninsulated wire tip was pressed on the silver foil while heating them up to 450 °C for a few seconds using a soldering iron. The counter electrode was connected by soldering where *ca.* 2 cm of the silver wire insulation was removed and wrapped around one end of the graphite rod.

The reference electrode was prepared by cleaning the stripped tip (*ca.* 2 mm) of a silver wire in concentrated nitric acid for 30 seconds and thereafter subsequently placed in an 1M aqueous solution of potassium chloride ($\geq$ 99.5 purity; Sigma Aldrich, Munich, Germany) for 30 minutes. During this process, a thin layer of silver chloride (AgCl) is formed, creating a micro Ag/AgCl reference electrode (Inzelt, 2013). The averaged potential of the micro Ag/AgCl reference electrode was determined to be $0.132 \pm 0.004$ V *vs.* a Ag/AgCl (3M KCl) reference electrode in 1M $KHCO_3$(aq). Subsequent potentials presented in the results and discussion section are provided *vs.* the micro Ag/AgCl electrode. The commercial electrode was specified with a potential of 0.210 V *vs.* normal hydrogen electrodes (NHE), resulting in a potential of $0.342 \pm 0.004$ V *vs.* NHE for the micro Ag/AgCl reference electrode.

The electrodes were arranged in a geometry as shown in Figure 1a and fixed using PTFE tape and shrinking tube. The distance between the center of the working electrode and the reference electrode was adapted to the height of the sensitive volume and the position of the working electrode inside the 5 mm tube was adjusted to match the center of the NMR coil. This minimizes the content of conductive material inside the NMR coil, thus reducing distortions of $B_0$ and interactions with $B_1$. Additionally, a short distance between the reference and the working electrode ensures a small uncompensated resistance of $5 \pm 2$ $\Omega$ and correspondingly a small iR drop for all electrochemical measurements.

The lead wires of the electrodes were passed through a drilled opening of an NMR tube cap. Cellulosenitrate glue (UHU HART, UHU, Bühl, Germany) was applied to the top of the tube cap and the protruding connection wires in order to fix the position of the electrodes inside the 5 mm tube and seal the drilled opening in the cap. Additionally, ethyl-cyanacrylate glue





(Loctite 406, Henkel, Düsseldorf, Germany) was applied on top after the cellulosenitrate glue hardened.

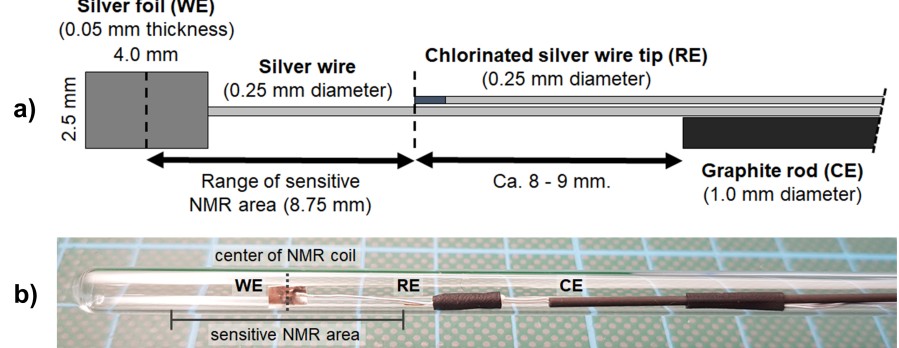

**Figure 1.** (a) Geometry and arrangement of the three-electrode *in operando* NMR setup. It consists of a silver foil working electrode (WE), a graphite rod counter electrode (CE) and a micro Ag/AgCl reference electrode (RE). The reference electrode was placed on the edge of the sensitive NMR area to minimize the amount of conductive material during NMR measurements while maintaining a small ohmic potential drop between working and reference electrode. (b) Photograph of the electrode setup inside a 5 mm tube.

## 2.2 Cell holder

The holder for the electrolysis cell is shown in Figure 2. The setup enables an easy and stable connection between the thin silver wires of the cell and the shielded coaxial cables of the potentiostat. Furthermore, it increases the structural stability of the cell by reducing the weight and strain as well as vibrations of the coaxial cables. The frame of the cell holder was 3D-printed using Acrylnitrile-Butadiene-Styrene (ABS) copolymer (Filamentworld, Neu-Ulm, Germany). For each electrode a non-magnetic SMA coaxial connector (model 23_SMA-50-0-13/111_NE, Huber+Suhner, Herisau, Switzerland) was fixed to the frame using

non-magnetic screws. For connection of the electrolysis cell, the silver wires were soldered to the connector pins. The bottom hole of the cell holder was adjusted to the diameter of the NMR tube and the tube cap. The 5 mm tube containing the electrolysis cell is mounted into the cell holder from the top opening and the cell is then fixed by tightly clamping the tube cap at the top end of the NMR tube into the bottom hole of the holder.

To stabilize the sample inside the magnet and to achive a mechanical separation of probe and cell, a dismounted turbine of a

magnet lift was fixed on top of the probe. A spinner was attached to the *in operando* cell, placed inside the turbine and inserted into the magnet.





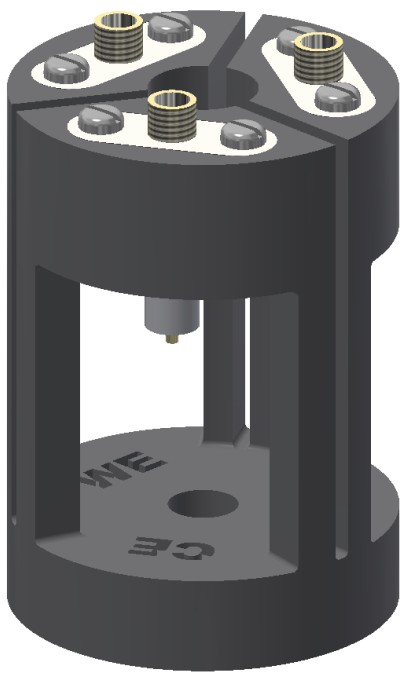

**Figure 2.** Cell holder consisting of the 3D-printed frame (black) and three SMA coaxial connectors (white and gold). The electrolysis cell is fixed inside the cell holder and the electrode wires are soldered to the pins of the SMA coaxial connectors.

## 2.3 Noise reduction assembly

The *in operando* cell was connected to the potentiostat using shielded coaxial cables with SMA connectors. The top opening of the magnet was closed with a copper plate containing RF feed-throughs for the cables to the potentiostat (NMR Service, Erfurt, Germany). Additionally, three low pass radio frequency filters (models SLP-5+, SLP-15+, SLP-30+, Mini Circuits, New York, USA) were connected to each cable in order to reduce RF noise from the potentiostat and environmental sources. A total of three low pass filters were connected to each cable. One low pass filter (SLP-5+, <5 MHz) was connected to the copper plate connections at the top of the magnet and two filters each (SLP-15+, <15 MHz & SLP-30+, <30 MHz) were placed directly at the potentiostat connections. As the connection cables and corresponding banana plugs attached to the potentiostat are unshielded, a silver cloth was wrapped around the unshielded cables. In addition, the body of the probe and the NMR magnet as well as the potentiostat were connected to a common ground. The shielding setup is shown in Figure 3.





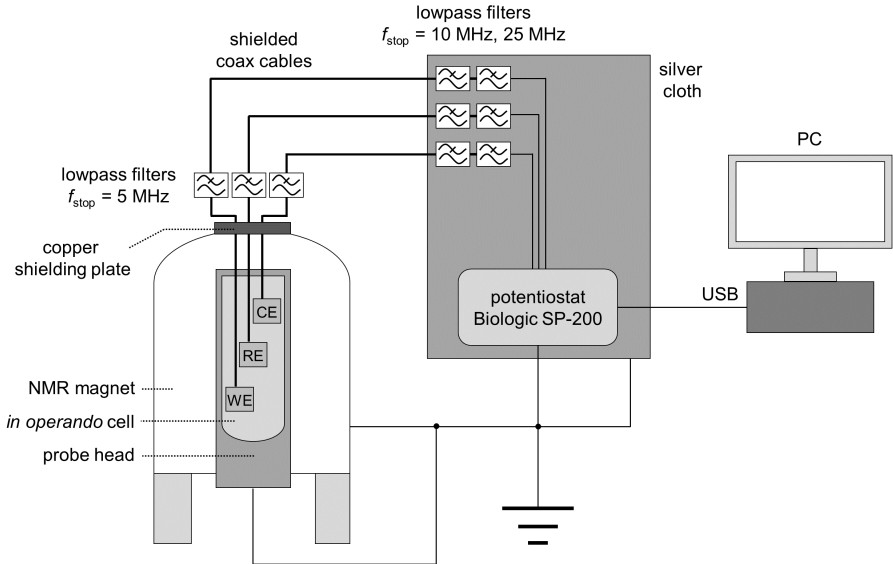

**Figure 3.** Schematic drawing of the *in operando* NMR electrolysis setup with shielding, RF filters and potentiostat.

### 2.4 $B_1$ field simulation and nutation behavior

The distortion of the $B_1$ field in the proximity of the metal electrode was numerically simulated using EMpro (Version 2020, Keysight Technologies). A square Helmholtz coil was designed to mimic a homogeneous RF field in the vicinity of the electrode consisting of two parallel square shaped wires with a distance of 0.5445 times the length for each side of the square.

An ideal conductor served as coil material, and both squares of the coil were driven synchronously by a current source. The silver electrode was placed in the center of the coil as shown in Figure 4d. The simulation was performed for three different

angles (0°, 45°, 90°) between the $B_1$ field and the electrode plane. Data points were acquired with a resolution of 0.4 mm. The complex magnetic field vectors of the simulated volume were exported by means of a Python script (Python 3.7, Python Software Foundation) for data processing.

Eddy currents are formed on the metal surface (Figure 4) caused by the oscillating $B_1$ field. In turn, the eddy currents induce a magnetic field that distorts the amplitude and phase of the excitation pulse. The distortion of the $B_1$ field strongly depends

on the angle between the electrode and the RF field (Romanenko et al., 2014; Britton, 2014). For a parallel configuration, *i.e.* at an angle of 0°, distortions of the $B_1$ field are minimized (Figure 4a). Correspondingly, there is only a small eddy current formation due to the minimal surface area remaining perpendicular to the $B_1$ field.

For a perpendicular (90°) orientation of the electrode (Figure 4c) the $B_1$ field showed major distortions which lead to a decrease in strength of the $B_1$ field in the proximity of the electrode surface. However, the $B_1$ field strength showed an

increase at the top and bottom edges of the electrode. At the side edges, the direction of the field, *i.e.* the phase of the $B_1$,



changed. At a distance of about 0.8 mm from the electrode surface, the strength of the RF distortions decreased to 1/e of the $B_1$ field.

Smaller distortions of $B_1$ were also observed for the 45° orientation of the electrode (Figure 4b), affecting mostly the direction of the field whereas the amplitude decreased at the surface.

Major distortions of $B_1$ phase were observed along the top and bottom edges of the electrode. The distortions decreased to 1/e of the $B_1$ field strength at a distance of *ca.* 0.6 mm from the electrode surface.

Concluding from the results of the simulation, an electrode orientation planar to $B_1$ can be considered optimal. For different angles, amplitude and phase of the $B_1$ field distortions depend on the spatial orientation. However, the distortions are mainly significant in the vast proximity of the electrode (0.6–0.8 mm from the surface). Therefore, the majority of the volume inside a

5 mm NMR tube is considered distortion free for the chosen electrode geometry and thus can be probed by NMR spectroscopy without additional measures. Thus, for the current setup a minute adjustment of the orientation is not necessary. It should be noted that the distortions of the $B_1$ field do not depend only on orientation but also on electrode size. Adjustments may be required for larger *in operando* electrolysis cells.

A nutation experiment of a $CO_2$ saturated 1M $KHCO_3$ solution in the *in operando* cell was performed using a Bruker Avance III HD spectrometer with a 14.1 T wide bore magnet (150.9 MHz RF frequency for $^{13}C$) and a broadband gradient probe (Bruker DiffBB). Data points were acquired in pulse length steps of 1 $\mu$s at a constant pulse power of 59 W. The nutation curve shows the integral of the $^{13}C$ $HCO_3^-$ due to its higher signal-to-noise ratio compared to the $CO_2$ signal (Figure 5a). The $B_1$ field distortions due to interactions with the metallic components of the cell result in a signal decay with a time constant of

$72 \pm 2.6$ $\mu$s. However, simple NMR experiments using 90° and 180° pulses remain manageable. The Fourier transform of the nutation curve (Figure 5b) shows a broadly distributed nutation frequency of the main component at 16.9 kHz with a full width at half maximum (FWHM) of 6.8 kHz. An additional component appears as a low frequency shoulder of the main component.

## 3 Methods and materials

A 1M aqueous solution of 98% $^{13}C$ enriched $KHCO_3$ (Sigma Aldrich, Munich, Germany) was used as electrolyte. The elec-

trolyte was pre-chilled inside a polyethylene vial in a 10 °C water bath. *Ca.* 1 mL of chilled electrolyte was filled into a 5 mm NMR tube and saturated with 99% $^{13}C$ enriched $CO_2$ (Cambridge Isotope Laboratories, Tewksbury, USA) by bubbling for 20 minutes at a temperature of 10 °C if not stated otherwise. The $CO_2$ was bubbled into the electrolyte using a 1/16 inch PEEK tube, and the flow rate was adjusted to *ca.* 0.3 mL/s. The three electrode setup was placed inside the 5 mm tube filled with $CO_2$ saturated electrolyte, ensuring that the contact between counter electrode and silver wire is not immersed in liquid. Prior

to sealing, the gas phase inside the tube was aerated with $^{13}C$ labeled $CO_2$ gas. All preparation steps were performed under ambient condition.

The electrochemical experiments were performed using a BioLogic SP-200 potentiostat (BioLogic Science Instruments, Seyssinet-Pariset, France) at a temperature of 10 °C, controlled by a surrounding water bath. The electrochemical perfor-



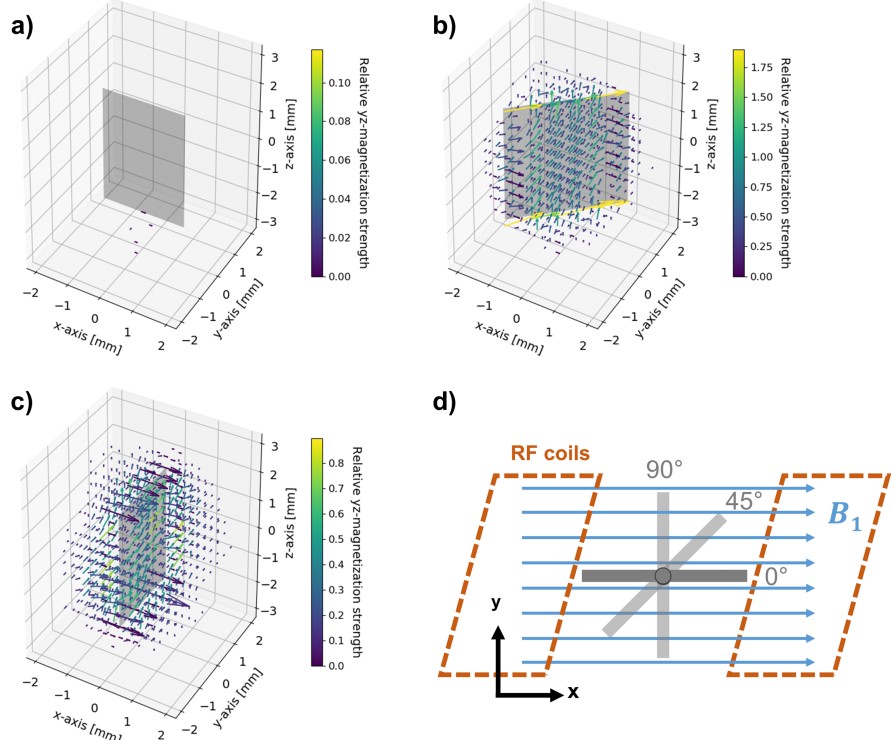

**Figure 4.** $B_1$ field simulation in proximity of the metal electrode for angles of 0° (a), 45° (b) and 90° (c) between the incoming RF field and the electrode, and geometry and arrangement of the metal electrode in relation to the $B_1$ field in the simulation (d). The incoming RF field points towards the positive x-axis. The vectors are the deviations in field strength and direction compared to the incoming RF field. For a better visibility of phase deviations, the vectors are color coded according to their relative field strength in y-direction compared to the incoming field strength, which possess only a component along the x-axis. Note that all figures have individual colorbar ranges and perspective. No distortion is present for a parallel (0°) orientation of RF field and electrode. The angled (45°) and perpendicular (90°) orientations cause major distortions in electrodes proximity, which diminish at a distance of 0.6–0.8 mm.

mance of the *in operando* cell was evaluated using chronopotentiometry (CP) at several current densities up to 4 mA/cm$^2$

for 15 minutes each and linear sweep voltammetry (LSV) in the range of -1.0 V to -2.5 V *vs.* Ag/AgCl (rate 10 mV/s) afterwards. In between the electrochemical experiments the system was allowed to relax for 5 minutes. For reference, an equivalent chronopotentiometry experiment was performed using a 1 cm$^2$ silver electrode and identical reference and counter electrodes. The reference chronopotentiometry experiment was performed in a cleaned glass beaker filled with 60 mL of aqueous $CO_2$ saturated 1M $KHCO_3$ electrolyte, which is denoted as bulk cell in later sections. In the bulk cell, working and counter electrode

were arranged in a parallel geometry with a distance between working and reference electrode identical to the *in operando* cell. The potential of the micro reference was measured *vs.* a commercial Ag/AgCl reference electrode with a double junction system and a 3M aqueous KCl bridge electrolyte. The measurement was performed in the electrolyte of the $CO_2$ electrolysis



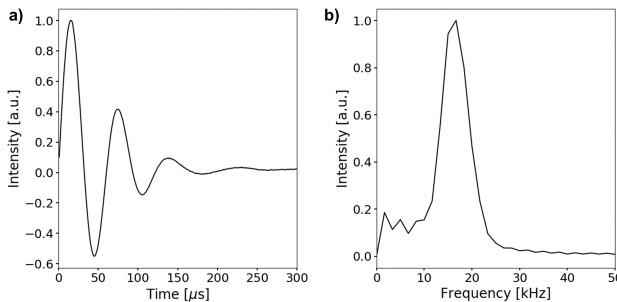

**Figure 5.** Nutation curve of the $^{13}C$ $HCO_3^-$ signal using the the *in operando* cell (a). Due to the $B_1$ field inhomogeneities, the signal decays with a time constant of $72 \pm 2.6$ $\mu s$. (b) shows the Fourier transform of the nutation curve. The main component of the magnetization nutates at a frequency of 16.9 kHz (15.5 $\mu s$ 90° pulse length), but is broadly distributed at a FWHM of 6.8 kHz. Additionally, there is a significant component at lower nutation frequency.

and the reference electrode potential was averaged over 10 minutes. Both reference electrodes were equilibrated for 10 minutes prior to the experiment.

All $^{13}C$ NMR measurements were performed using a Bruker Avance III HD spectrometer with a 14.1 T widebore magnet (150.9 MHz RF frequency for $^{13}C$) and a broadband gradient probe (Bruker DiffBB). Spectra were post processed applying a 1 Hz line broadening. NMR experiments were performed at a temperature of 10 °C if not stated otherwise. Concentrations of the carbon species in the $CO_2$ saturated electrolyte with $CO_2$ atmosphere were determined in a sealed NMR tube using sodium trimethylsilylpropanesulfonate (DSS) (Sigma Aldrich, Munich, Germany) as reference ($c$(DSS) = 61.62 mmol/L) and

a $^1H$ WALTZ-16 sequence for decoupling (Shaka et al., 1983; Tenailleau and Akoka, 2007). The chemical shift scale of all future $^{13}C$ spectra was referenced using the frequency offset of DSS in this experiment. DSS was not included for *in operando* experiments, because the organic salt may change electrochemical behavior. The $CO_2$ saturated electrolyte was examined using $T_1$ and $T_2$ relaxation and exchange time measurements. Relaxation time constants were determined using a saturation recovery pulse sequence for the determination of $T_1$ and a Carr–Purcell–Meiboom–Gill (CPMG) pulse sequence for $T_2$ measurements

(Carr and Purcell, 1954; Meiboom and Gill, 1958). The exchange time between $HCO_3^-$ and solvated $CO_2$ was assessed by a 1D EXSY sequence, which uses a shaped Gauss pulse for the selective inversion (100 Hz bandwidth) of the bicarbonate resonance at 160.7 ppm. The center frequency of the selective inversion pulse was adjusted in case of a $HCO_3^-$ frequency shift.

The exchange time constant $T_{exc}$ was determined by fitting the evolution of the $CO_2$ signal integral $I(CO_2)$ as a function of the mixing time $\tau_m$ to

$$I(CO_2) = I_0(CO_2)\left\{1 - 2\left[\exp\left(-\frac{\tau_m}{T_{exc} + T_1}\right) - \exp\left(-\frac{\tau_m}{T_1}\right)\right]\right\}, \tag{1}$$

where $I_0$ is the signal integral at $\tau_m = 0$. This simplified fitting equation is valid under the conditions that the concentration of bicarbonate substantially exceeds the $CO_2$ concentration, and that both species have similar longitudinal relaxation times $T_1$.





# 4   Results and discussion

## 4.1   Electrochemical performance of the *in operando* electrolysis cell

The time dependent potential curves for the chronopotentiometry measurements are shown in Figure 6a. The potentials ob-

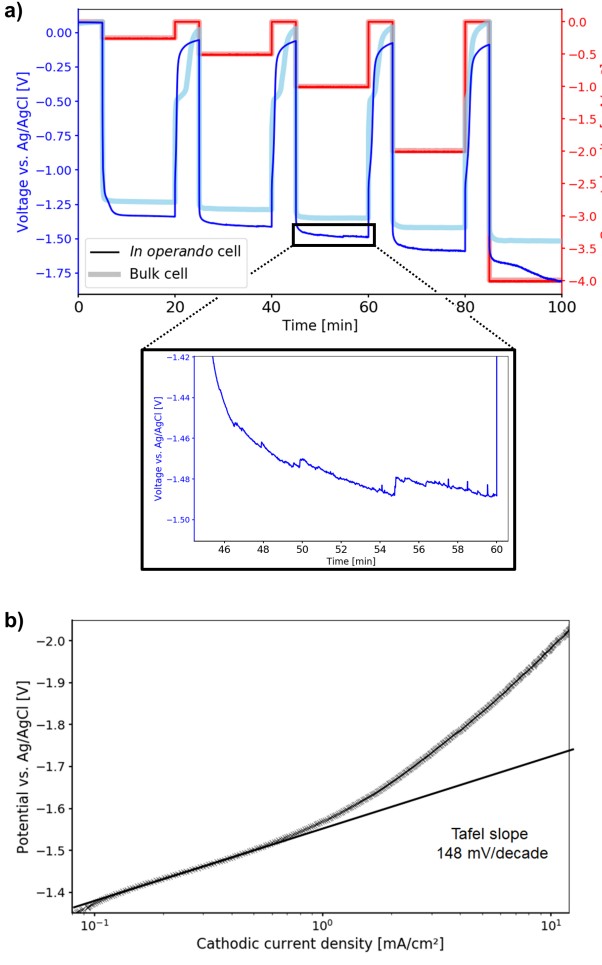

**Figure 6.** Time depended potential curves during the chronopotentiometry measurement (a). Electrolytic reduction of $CO_2$ starts at -1.33 V *vs.* Ag/AgCl for the *in operando* cell. Compared to the bulk cell, higher overpotentials are observed. Starting at 1 mA/cm², oscillations and increased noise appear which are caused by stuck product gas bubbles. Tafel plot of the electrolytic $CO_2$ reduction in the *in operando* electrolysis cell (b). The tafel slope was determined in the low current density region as 148 mV per decade, resulting in a transfer coefficient of 0.38 at 10 °C.


served for both the *in operando* and the bulk cell are within the range reported in literature, as the values strongly depend on the properties of the catalyst and the electrolysis cell. $CO_2$ electrolysis starts at -1.33 V *vs.* Ag/AgCl for the *in operando* cell





and -1.23 V *vs.* Ag/AgCl for the bulk cell. However, three kinds of deviations in the potential curves for the *in operando* cell compared to the bulk cell were identified. First, higher overpotentials are observed. Second, the *in operando* cell takes longer

to equilibrate both when a current is applied and when it is switched off. Third, increasing oscillations of the potential and noise are observed, starting at -1 mA/cm$^2$.

The non-parallel geometry of the electrodes in the *in operando* cell may be causing the first two deviations. In a parallel geometry as employed in the bulk cell, the distance of the working electrode to the counter electrode is approximately constant across the whole WE surface. Thus, the iR drop is constant across the electrode and the surface potential is uniform. However,

in a non-parallel geometry, there is a spatially dependent iR drop between working and counter electrode, which leads to non-uniform potential distribution across the electrode surface. As a result, electrolysis preferably takes place at the closest region between working and counter electrode, where the iR drop is minimal. The $CO_2$ concentration in that region decreases during electrolysis, and thus the concentration overpotential increases. At a point in time, the concentration overpotential exceeds the increase in iR drop for a more distant region, which results in the electrolysis taking place at that location.

For the electrode setup used in the *in operando* cell, the region of the working electrode closest to the counter electrode is the edge of the silver sheet. For a current density of -0.25 mA/cm$^2$, the electrolytic $CO_2$ reduction at the silver foil edge takes place at the same overpotential compared to the bulk cell . However, the electrolysis at the electrode edge is not sustainable because of the small area and therefore the diffusive $CO_2$ transport becomes limited. As a result, the $CO_2$ concentration is depleted after 2 minutes of electrolysis. At this point, $CO_2$ reduction takes place at the next-nearest region to the counter electrode,

which is a portion of the silver foil plane, where $CO_2$ can be more readily regenerated by diffusion. However, the iR drop at the silver foil plane are increased compared to the edge, and thus the potential decreases.

With increasing current density, $CO_2$ is consumed at an increasing rate. Therefore growing portions of the silver electrode surface with an increasing distance to the counter electrode participate in the electrolytic reduction reaction. This results in a rising iR drop, and the overpotential increases further compared to the bulk cell. This effect may also lead to the unstable

potential of the *in operando* cell at a current density of -4 mA/cm$^2$. It is important to separate this effect from the expected increase in concentration overpotential with increasing current density, which was also observed for the bulk cell.

The oscillations and increased noise observed for the potential curve of the *in operando* cell at higher current densities are caused by the formation of gaseous products, *i.e.* CO and $H_2$, in the confined cell geometry. The gas bubbles tend to stick to the glass walls, electrodes or the connection wires, until they grow to a sufficient size to rise to the top. Diameters of up to 1/3 of

the size of the electrode surface were observed for the gas bubbles. As a result, the bubbles blocked significant fractions of the electrodes from participating in the electrolysis reaction, thus affecting the electrochemical measurements. For the bulk cell the larger cell geometry didn't promote the gas bubbles to stick, and the larger electrode size was not affected by the comparatively smaller gas bubbles.

For the *in operando* cell, the Tafel slope was determined as 148 mV per decade from current-voltage (IV) curves of the

LSV experiments in the low current density region as shown in Figure 6b. In this region no mass transport limitations for the electrolytic reduction of $CO_2$ were found. The slope of the Tafel-plot translates to a charge transfer coefficient of $\alpha = 0.38$. From literature, values for the Tafel slope can range from 130 mV to 140 mV per decade, resulting in charge transfer coefficients





of 0.41 - 0.45 for comparable systems under room temperature. (Hori, 2008; Hori et al., 1987; Endrődi et al., 2017; Lu et al., 2014; Hatsukade et al., 2014; Hsieh et al., 2015). The small discrepancies between measured and literature values may

originate from the lower temperature at which the shown experiments were performed, resulting in lower thermal energy for the activation of processes and lower diffusion rates.

Overall, the *in operando* cell shows a comparable performance to a bulk electrolysis cell in the low current density range, *i.e.* below -1 mA/cm$^2$. Due to the non-parallel cell geometry there is a spatially dependent iR drop distribution across the working electrode surface, which is unfavorable for precise electrolysis experiments. To minimize this effect, an electrode with a small

geometry was chosen. Nonetheless, the potential is unstable at higher current densities, and electrochemical measurements may be distorted by gas bubbles stuck in the confined glass tube.

### 4.2   NMR Evaluation of the *in operando* electrolysis setup

The $^{13}$C spectrum of $CO_2$ saturated electrolyte is shown in Figure 7a. The two signals in the spectra are assigned to $HCO_3^-$ at

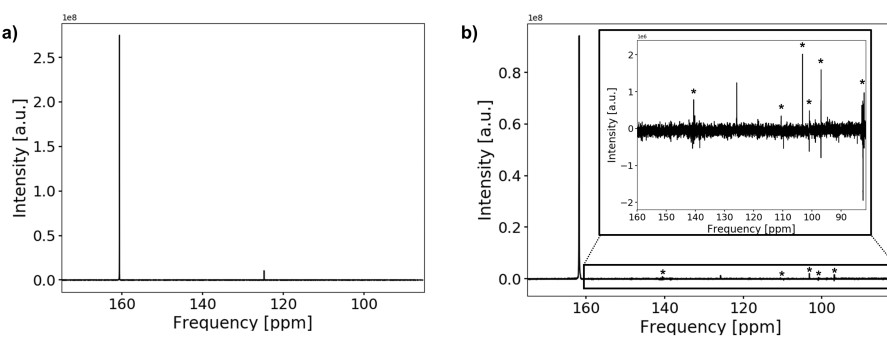

**Figure 7.** $^{13}$C spectrum of the $CO_2$ saturated electrolyte without (a) and with (b) electrodes. The experiment with electrodes includes connection cables and a powered potentiostat, but no shielding. The peak positions of bicarbonate and solvated carbon dioxide are at 160.7 ppm and 124.7 ppm, respectively. Peak positions are shifted downfield by about 1.1 ppm when the conductive components are introduced. The spectrum in b) suffers from increased noise as well as from external RF signals, which are comparable in intensity to the $CO_2$ signal. The external RF signals have been marked with (*).

160.7 ppm and solvated $CO_2$ at 124.7 ppm (Holmes et al., 1998; Mani et al., 2006; Seravalli and Ragsdale, 2008; Abbott et al.,

1982; Liger-Belair et al., 2003). The low solubility of carbon dioxide in water causes a weaker signal compared to bicarbonate. No signal of gaseous carbon dioxide could be observed, as the gas phase inside the NMR tube is outside of the sensitive volume of the RF coil. For a measured pH of $8.2 \pm 0.15$ after $CO_2$ aeration of the electrolyte, about one percent of soluted carbonate is expected. However, due to the fast exchange of $HCO_3^-$ and $CO_3^{2-}$ signals for both species coalesced into one peak.

The concentrations of the carbon species in the $CO_2$ saturated electrolyte are shown in Table 1. During aeration of the elec-

trolyte with $CO_2$, the concentration of bicarbonate ($HCO_3^-$) increased by a factor of two. The measured concentration of $CO_2$ is within the uncertainty limits of the theoretical equilibrium concentration in aqueous solution ($c_{1013 \text{ hPa, } 10 \text{ °C}}$ = 52.7 mmol/L)





(Sander, 2015; Dodds et al., 1956). This uncertainty is caused by a combination of a weak signal-to-noise ratio of the DSS
methyl group reference signal and imperfections of the temperature control ($\leq$ 1°C).

**Table 1.** Concentrations of carbon species in $CO_2$ saturated electrolyte at 10 °C and 1013 hPa $CO_2$ pressure.

| Carbon species | Concentration [mol/L] |
|---|---|
| $HCO_3^-$ | $1.82 \pm 0.14$ |
| $CO_2$ | $(55.0 \pm 4.4) \cdot 10^{-3}$ |
| Total carbon | $1.87 \pm 0.14$ |

As the solubility of $CO_2$ in water is low, the $^{13}C$ signal of $CO_2$ is weak even when using $^{13}C$ labeled and fully saturated

samples. It is therefore essential to optimize the signal-to-noise ratio before performing further experiments. To investigate
the effect of the *in operando* setup on $^{13}C$ spectra, the signal-to-noise ratio of the $HCO_3^-$ signal was determined for different
conditions (Table 2). Decreasing the temperature from 22 °C to 10 °C significantly improved the signal-to-noise ratio. As

**Table 2.** signal-to-noise ratio of the $^{13}C$ $HCO_3^-$ signal under different conditions.

| Temperature [°C] | Conditions | signal-to-noise ratio [-] |
|---|---|---|
| 22 | No conductive material | 1247 |
| | *In operando* cell | 397 |
| | *In operando* cell and connection cables | 290 |
| 10 | No conductive material | 1994 |
| | *In operando* cell and connection cables | 399 |
| | Full *in operando* electrolysis setup with shielding | 2510 |

$CO_2$ possess a higher solubility at lower temperatures (*cf.* $c_{1013\text{ hPa, 22 °C}}$ = 38.0 mmol/L) (Sander, 2015; Dodds et al., 1956),
the signal-to-noise ratio of the $CO_2$ signal increased by a factor of *ca.* 1.4. The equilibrium constant for the $CO_2/HCO_3^-$

equilibrium changed by only 1% due to the decrease in temperature. Therefore an increase in $CO_2$ concentration causes
similar increases of $HCO_3^-$ in solution. The decrease of 12 °C also increases the equilibrium magnetization and thus reduces
thermal noise what led to an increase in the signal-to-noise ratio by about 6%.

After introducing the electrodes into the magnet the signal positions shifted downfield by 1.1 ppm and line widths become
significantly broader but could be reduced down to about 1 Hz after shimming except for a downfield shoulder. The signal-to-

noise ratio of the spectrum was reduced significantly by 68%. As the concentration of the carbon species remained unchanged,
the decrease in the signal-to-noise is a combined effect of increased noise levels and a reduced quality factor, $Q$, of the NMR





**Table 3.** Relaxation and exchange times for bicarbonate and carbon dioxide without conductive materials, with electrolysis cell and full electrolysis setup at 10 °C. The full electrolysis setup included the *in operando* cell, connection cables, a powered potentiostat and shielding equipment. For measurements using the electrolysis setup no electrochemical experiments were conducted.

|  |  | Without conductive materials | With electrolysis cell | With full electrolysis setup |
|---|---|---|---|---|
| $HCO_3^-$ | $T_1$ [s] | $18.59 \pm 0.08$ | $18.56 \pm 0.05$ | $12.25 \pm 0.02$ |
|  | $T_2$ [s] | $2.04 \pm 0.00$ | $1.40 \pm 0.00$ | $0.97 \pm 0.00$ |
| $CO_2$ | $T_1$ [s] | $20.15 \pm 0.59$ | $19.55 \pm 0.42$ | $13.99 \pm 0.65$ |
|  | $T_2$ [s] | $4.15 \pm 0.11$ | $2.03 \pm 0.05$ | $2.66 \pm 0.21$ |
|  | $T_{exc}$ [s] | $5.23 \pm 0.18$ | $3.31 \pm 0.25$ | $3.79 \pm 0.37$ |

circuit caused by the conductive components. As shown by Figure 7b, the main contribution is the introduction of external RF noise due to the metallic components and cables acting as radio antenna. Coherent external RF noise in the frequency range of $^{13}C$ NMR at 14.1 T (150.9 MHz) is caused by mobile radio communication (Bundesnetzagentur, 2019). Introducing

additional connections to the setup as well as connecting the cell directly to a powered potentiostat further decreased the signal-to-noise ratio despite using shielded coaxial cables. A highly shielded setup as described in in Figure 3 is therefore necessary to decrease RF noise originating from external sources in order to obtain signal-to-noise ratios comparable to experiments without conductive materials. Using just single elements of the shielding setup, *i.e.* only the copper plate for the top opening of the magnet, the silver cloth, the common ground or the filters, does not restore the signal-to-noise ratio to original values.

First, as a reference for the $CO_2$ saturated electrolyte, longitudinal relaxation times and exchange rates were determined using a standard NMR tube without the electrolysis setup. In a second step the electrodes and leads were present but not connected. In the final step dataa was collected with the full electrolysis setup shown in Figure 3. The results of all experiments are summarized in Table 3. The errors of the $CO_2$ rates are caused by a low signal-to-noise ratio of the carbon dioxide signal. Both species in the $CO_2$ saturated electrolyte show similar values of longitudinal relaxation rates. The longitidunal relaxation

rates for $HCO_3^-$ and $CO_2$ are unchanged within error boundaries after insertion of the electrodes. On the other hand, the exchange time constant of the chemical equilibrium between $CO_2$ and $HCO_3^-$ decreased from 4.55 s to 3.10 s. The decreased exchange time affects the transverse relaxation time constant and as a result decreases $T_2$ for both $HCO_3^-$ and $CO_2$. The change in exchange time is assumed to be caused by an interaction of $HCO_3^-$ with the polarizable silver metal electrode surface. The interaction of ions with metal surfaces by induction of dipoles is reported in literature to extend up to 1 nm from the metal

(Bonzel, 1988; Mendonca et al., 2012; Seitz-Beywl et al., 1992). To effect the bulk solution, the exchange between free ions and $HCO_3^-$-metal complexes has to be sufficiently fast in addition to a high catalytic activity of the $HCO_3^-$/metal surface. On the one hand, the positively charged metal surface acts as a catalytic center for the $CO_2$/$HCO_3^-$ equilibrium reaction by



stabilization of intermediate compounds and thus decreasing the exchange time. Catalysts of the $CO_2/HCO_3^-$ equilibrium are well known and important for biological systems in form of the carbonic anhydrase enzymes. Carbonic anhydrase functions

similarly by stabilization of the negatively charged oxygen atoms by metal cations during the $CO_2/HCO_3^-$ exchange reaction and increases the reaction rate by six to seven orders of magnitude (Lindskog, 1997; Grisham and Garrett, 2010). On the other hand, the fast exchange between surface and bulk species may be caused the RF pulses which induce eddy current in the silver metal. These eddy currents increase the temperature of the electrolyte in the electrode vicinity what in turn creates a convection flow and causing a mixing of the solution.

During the measurements employing the full *in operando* electrolysis setup, the cell was connected to a potentiostat. The potentiostat was powered on, but no electrochemical experiment was conducted. Therefore the cell operates in a open circuit voltage (OCV) mode, where no current flows between the electrodes, but the voltage between the electrodes is continuously measured by the potentiostat. Similar to the experiments where the electrolysis cell has been disconnected, the exchange time between carbon dioxide and bicarbonate remains unchanged within error boundaries. However, the longitudinal $^{13}$C

relaxation time constant for bicarbonate and $CO_2$ and the transverse relaxation time constant for bicarbonate were found to be smaller. As the experimental setup inside the sensitive volume remained unchanged, the leads and filters as well as the potentiostat may be the driving forces for the increased relaxation rates. The continuous voltage measurement of the powered potentiostat should not have a considerable influence, since the potentiostat input is terminated with high impedance. However, all voltage measurements cause a minuscule current flow between the cell and the potentiostat, thus double layer formation

and mobility of the electroactive species may be affected. Nevertheless, it is improbable that increased stochastic fluctuations of magnetic fields originating from the potentiostat are causing the increased relaxation rate. While powering on the potentiostat causes increased RF noise in the NMR experiment, these fluctuations are successfully removed by the filters described in subsection 2.3, therefore such a drastic effect on relaxation is not expected.

A more probable source for the altered relaxation behavior is the changed capacity of the electrode assembly. Cables and

filters can contain or act as capacitors, which changes the capability of the setup to dissipate or provide electrons at the electrodes. As OCV is an electrostatic mode of operation, the assembly may act as a additional power supply and thus affect double layer formation. This in turn may affect the whole electrolyte, *e.g.* via changing the equilibrium between the ionic species, which may lead to an altered pH of the system. This is known to sensitively affect relaxation properties for aqueous carbonate solutions (Moret et al., 2013). While a detailed analysis of these processes is outside the scope of the current study,

it highlights the sensitivity of $^{13}$C NMR to investigate fundamental processes occurring during $CO_2$ electrolysis, thereby justifying the efforts necessary to achieve sufficient sensitivity and resolution for *in operando* experiments. At the same time, this also demonstrates the importance for a proper designed electrolysis setup to avoid unwanted side effects.

The results also show that any sort of measurement setup may affect an electrochemical system. While the *in operando* electrolysis setup does not disturb the NMR measurements, it can affect the state of the electrodes and thus their interaction

with the $CO_2$ saturated electrolyte. However, the NMR measurements of the unconnected and connected electrolysis setup show that even the equipment which is imperative for electrochemical testing can affect the equilibrium state of the electrolysis. This is particularly pronounced at low current densities as OCV.





### 4.3 *In operando* NMR of the OCV evolution

The $^{13}$C NMR spectra of the aqueous $HCO_3^-$/$CO_2$ sample during OCV and the potential between working and reference electrode are shown as a function of time in Figure 8. The current density between working electrode and counter electrodes

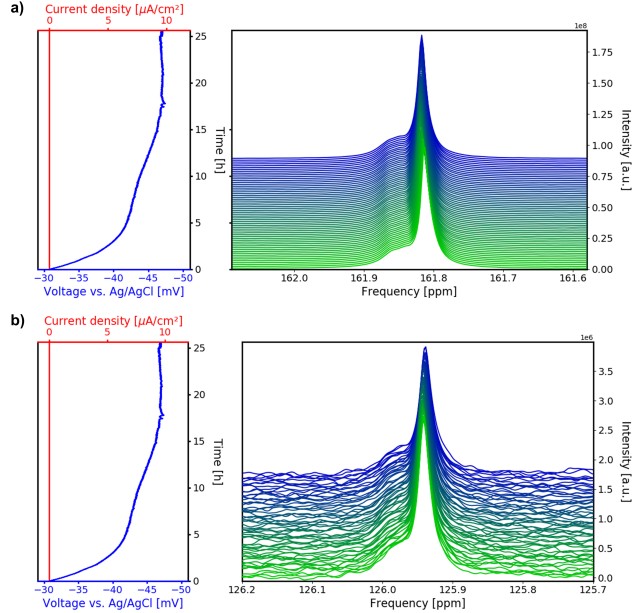

**Figure 8.** Time evolution of the $^{13}$C signals for $HCO_3^-$ (a) and $CO_2$ (b) during the OCV stage versus electrochemical potential and current density. In each sub panel the time dependent potential and current density is shown on the left, and the corresponding spectra are shown on the right. After a relaxation period, the potential remains at a stable potential of 47 mV.


remains fixed at 0 mA/cm$^2$ during measurements. During the first 5 hours of the experiment the potential drops from -31 mV to -42 mV *vs.* Ag/AgCl. After 12 hours the potential plateaued at -45 mV *vs.* Ag/AgCl and approaches equilibrium of -47 mV *vs.* Ag/AgCl after 17 hours.

The$^{13}$C NMR peaks of $HCO_3^-$ and the $CO_2$ are at the same initial position compared to the reference measurements. A
narrow main peak with broader shoulder persisted throughout the OCV stage. Fitting both signals with a Lorantzian line shape, a peak separation of 0.04 ppm (6.1 Hz) is obtained at 14.1 T. The shoulder peak is assumed to be caused by $B_0$ field distortions in the proximity of the working electrode, which cannot be easily corrected by shimming.

The $HCO_3^-$ signal drifted downfield about 7 ppb during the first 5 hours (Figure 8a), whereas the $CO_2$ signal shifted by 2 ppb (Figure 8b). Therefore, the evolution of the $HCO_3^-$ signal position is not caused by a magnet drift. The integral of the$HCO_3^-$
signal (Figure 9a) drops by 1%. The $HCO_3^-$ signal integral increases *ca.* 1% compared to the initial intensity. The evolution of the $HCO_3^-$ signal position and the intensity imply an evolution of the $CO_2$/$HCO_3^-$/$CO_3^{2-}$ equilibrium since a higher chemical shift is associated with an increase of the $CO_3^{2-}$ concentration (Abbott et al., 1982).





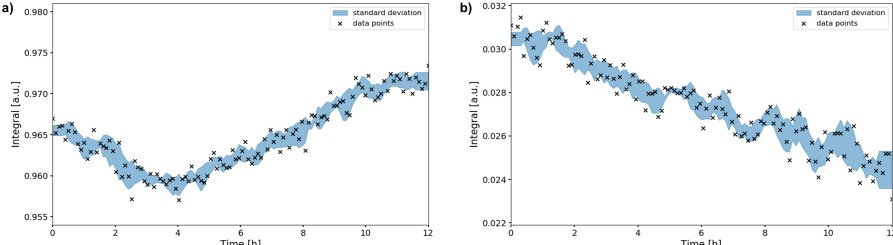

**Figure 9.** Time evolution of the $HCO_3^-$ (a) and $CO_2$ (b) signal integrals during the OCV stage. The integrals were normalized to their maximum values during the *in operando* experiment. Error boundaries are shown in blue. The $HCO_3^-$ signal fluctuates within the 1% range, while the $CO_2$ signal decreases significantly in intensity over the 12 hour period, down to 78% of its maximum value.

The intensity of the $CO_2$ peak continuously decreases during the OCV stage. After 12 hours the $CO_2$ signal integral decreased to 78% of the initial value (Figure 9b). Using 55.0 mmol/L as the initial concentration, as was determined in the reference experiment, this equals a concentration of 42.9 mmol/L. After 25.6 hours of OCV, the $CO_2$ signal intensity decreased to 62% (34.1 mmol/L). This behavior is unexpected as no $CO_2$ has been converted by electrolytic processes during the OCV stage.

Leakage of such a large amount of $CO_2$ gas during the NMR experiment is unlikely. Permeation of carbon dioxide through the polypropylene tube cap or the glue used for sealing can be excluded, as the $CO_2$ gas permeability for these materials is low (Hasbullah et al., 2000). Furthermore, any $CO_2$ loss should be compensated by the $CO_2$/$HCO_3^-$ equilibrium reaction, thus decreasing the $HCO_3^-$ concentration. However, no sustained decrease in $HCO_3^-$ concentration is observed. Furthermore, the total amount of all carbon species is unchanged after 12 hours of OCV, and therefore no $CO_2$ was lost to the environment.

It can be therefore concluded that initially, the $CO_2$ saturated electrolyte is not at equilibrium, which can be confirmed by the following considerations. Directly after preparation, the pH value of the electrolyte was $8.2 \pm 0.15$. Given a total concentration of 1.87 mol/L for all carbon species, the equilibrium concentration of solvated $CO_2$ at that pH value is 33.7 mmol/L. Thus the initial $CO_2$ concentration of 55.0 mmol/L is above the equilibrium value. The $CO_2$/$HCO_3^-$ system reaches equilibrium at the end of the OCV experiment, when the $CO_2$ concentration reaches 34.1 mmol/L

All changes in the $HCO_3^-$ and $CO_2$ signal integrals and the $HCO_3^-$ signal position occur in accordance to the variations of the potential during OCV. Changes in $HCO_3^-$ and $CO_2$ signal are associated with a shift in equilibrium, what indicates an shift in the electrochemical potential during the OCV stage caused by an evolution of the $CO_2$/$HCO_3^-$/$CO_3^{2-}$ equilibrium.

All relaxation and exchange times of $CO_2$ and $HCO_3^-$ during the OCV stage are given in Table 4. Compared to the test of the *in operando* cell, the exchange time slightly decreased after the system approached electrochemical equilibrium, which corresponds to the increased $HCO_3^-$ concentration. The decomposition rate of bicarbonate, *i.e.* the inverse of the exchange time, is linearly proportional to the $HCO_3^-$ concentration. The decreased exchange time between $CO_2$ and $HCO_3^-$ affects the transverse relaxation process and decreases $T_2$ time constants. Longitudinal relaxation time constants decreased only slightly as result of the change in equilibrium and are overall comparable to the test of the *in operando* setup.





**Table 4.** Relaxation and exchange times for bicarbonate and carbon dioxide during OCV.

|  | $T_1$ [s] | $T_2$ [s] | $T_{\mathrm{exc}}$ [s] |
|---|---|---|---|
| $HCO_3^-$ | $11.80 \pm 0.03$ | $0.78 \pm 0.01$ | |
| $CO_2$ | $13.18 \pm 0.71$ | $2.15 \pm 0.25$ | $2.65 \pm 0.28$ |

## 5 Conclusions

In this study, a setup for the in *in operando* NMR study of the electrochemical $CO_2$ reduction was developed, which was specifically designed to observe molecular dynamics changes in proximity to the working electrode. A key feature of the *in*
*operando* setup is the suppression of noise and external radio frequency signals that were introduced by conductive materials, enabling the observation of low concentration species. Relaxation and exchange experiments provide a sensitive probe for the interaction of ionic species and metal electrodes under different electrochemical conditions. The results of those experiments indicated that the electrochemical measurement equipment itself may affect a reaction and molecular dynamics. Finally, *in operando* NMR was used to monitor an aqueous $CO_2/HCO_3^-$ system for electrolytic $CO_2$ reduction at open circuit voltage. It
was revealed, that (electro-)chemical equilibrium in solution evolves for considerable time after sample preparation.

*Author contributions.* Each author contributed to this work as follows. S. Jovanovic developed the *in operando* cell and setup with assistance of P. Schleker, P. Jakes and J. Granwehr. NMR experiments were performed by S. Jovanovic. $B_1$ field simulations were performed by M. Streun. Data analysis and interpretation were performed by S. Jovanovic in collaboration with P.Schleker, P. Jakes, S. Merz, J. Granwehr and R.-A. Eichel. The manuscript was written in collaboration by S. Jovanovic, S. Merz, J. Granwehr and R.-A. Eichel. All authors have read
and agreed to the manuscript.

*Competing interests.* The authors declare no conflicts of interest.

*Acknowledgements.* The author gratefully acknowledges funding by the German Federal Ministry of Education and Research (BMBF) within the Kopernikus Project P2X: Flexible use of renewable resources – research, validation and implementation of 'Power-to-X' concepts, by the Deutsche Forschungsgemeinschaft (DFG, German Research Foundation) under Germany´s Excellence Strategy – Cluster of Excellence
2186 „The Fuel Science Center" – ID: 390919832, and project SABLE (grant 03EK3543) for the 600 MHz NMR spectrometer.



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
