# Peer review of "An electrochemical cell for *in operando* 13C NMR investigations of carbon dioxide/carbonate processes in aqueous solution"

_Magnetic Resonance, 2021_

## Author Response (AR1)

The authors thank both reviewers for their remarks, which helped us improve the manuscript and communicate our findings. We hope we addressed all comments to the satisfaction of the reviewers.

**Reviewer 1**

This is clearly a very challenging subject, to quantitatively characterize the evolution of 13 C NMR signals in the presence of conductive metals, electrochemical processes and various sources of noise and artifacts.

I believe, there are many unknowns left in this study, and questions raised by the authors in the manuscript need to be addressed and verified by them in a systematic manner., e.g., the relaxation measurement inconsistencies, unexpected CO2 signal decay during OCV and effects of bubbles on homogeneity and susceptibility.

1. Relaxation measurement inconsistencies: The corresponding paragraph was edited for clarity (341 ff.). Therein, we offer insights for the origin of the relaxation effects and thoroughly discuss the changes in relaxation times. Due to the step-by-step approach we followed during our studies, we therefore do not consider the changing relaxation times to be inconsistencies.
2. Unexpected $CO_2$ signal decay: The corresponding paragraph was edited for clarity (407 ff.). The monitoring of the $HCO_3^-$ and $CO_2$ signal during OCV were used as a validation of the in operando setup. To our knowledge no other study in literature exists where the $HCO_3^-$ / $CO_2$ equilibrium in an electrochemical cell is monitored for a prolonged time. We discussed the evolution of the signals thoroughly and concluded that it must originate from a shift in $HCO_3^-$/ $CO_2$ equilibrium as other sources could be systematically excluded.
3. Effect of bubbles on homogeneity and susceptibility: No bubbles occurred during the NMR experiments as no electrolytic reaction was taking place. The bubbles were only present during the experiments for electrochemical testing and we showed that bubble production can be minimized by utilizing low current densities. Therefore we do not consider bubble production an issue inside the scope of the manuscript. We added a short statement at the beginning of the in operando section for clarity (394).

If authors promote this cell design as an advantageous one, there should be a clear evidence of that in terms of quantitative data. I suggest a major revision, which should include troubleshooting addressing the questions raised by the authors, may be with the use of simplified cell design.

In our manuscript, we highlight the relative ease of construction as a highlight, which enables the adaptation of the cell in other NMR labs. This is an advantage which cannot be quantified in terms of data, but is still important nonetheless. However, we also present the benefits of the shielding setup, which is part of the equipment for the full electrolysis setup, as it increases the signal-to-noise ratio during NMR experiments by a factor of 6.

Looking at the provided references in literature regarding *in operando* cell designs, we do not see a possibility to simplify the cell design further. Working and counter electrodes are required, and it is highly recommended to employ a reference electrode. Shrinking of the working electrode surface both increases current densities and reduces the catalytic surface. The counter electrode does not influence the NMR experiments due to its position far outside of the sensitive area.

Lastly, we further addressed the questions regarding $B_1$ distortions by the metallic components of the cell by performing nutation experiments with variable electrode orientations (180 ff.)

Assessment criteria during the full review:

The English needs to be improved quite a bit. A number of errors were noted and paper requires extensive proofreading… e.g. "Lorantzian line shape" – line 365;

We improved the English throughout the manuscript.

What is ppb (line 368 ), did authors mean "ppm" or "part per billion" ? It is hard to see that from the spectra.

In the manuscript ppb was used for parts per billion, i.e. 1/1000 ppm. We changed from units in ppb to ppm for clarity.

The units of concentration are M, or mM… (mmol/L would be mM, line 387);

For units of concentration, both M and mol/L were used in the manuscript. Units of mol/L were changed to M for consistency.

Line 381: to use "was observed".

We fixed the grammar.

Line 13: "dynamics of the bicarbonate electrolyte changes"

We corrected the spelling.

Line 35: "chemical and reaction analysis" … not clear wording... is it "chemical reaction analysis" ?

We clarified the meaning of the sentence.

Line40: "Several experiment setups … were published"

We corrected the grammar.

Line 80: references should appear in chronological or alphabetic order.

We reorganized references in chronological order.

Line91 to use: "A three-electrode electrolysis cell that fits a standard 5 mm NMR tube has been build."

We fixed the sentence structure.

Define "iR" in the text.

We defined and explained the term iR drop.

Line 275: what is "small geometry" ? "compact design " ?

We clarified and expanded this section of the text.

Line 298: "After introducing the …widths became.."

We corrected the grammar.

Fig.4 needs to be more clearly presented.

Figure 4 was presented more clearly by focusing on the volume close to the electrode, i.e. by zooming into the volume of interest, and decreasing the density of distortion vectors.

**Reviewer 2**

This is a manuscript describing wonderful and detailed work showing in operando measurements of the carbon dioxide / carbonate interconversion on an electrode. It is a challenging problem, and it is indeed interesting to see that this reaction can be monitored in this way. I have the following minor comments:

(1) The discussion about the following is really quite unclear. I could not make sense of this from reading the text and looking at the figure: was the sample rotated or not, why? Do you really need mechanical separation? Why?

"To stabilize the sample inside the magnet and to achieve a mechanical separation of probe and cell, a dismounted turbine of a 135 magnet lift was fixed on top of the probe. A spinner was attached to the *in operando* cell, placed inside the turbine and inserted"

We clarified the procedure in the manuscript in accordance with our first comment on that question (133 ff.).

(2) The discussion in relationship to Fig. 4 is nice, but I would suggest to add the following references, which have also shown the orientation effects quite nicely:

https://pubmed.ncbi.nlm.nih.gov/25036296/

https://pubmed.ncbi.nlm.nih.gov/29960130/

This paper also discusses the orientation effect, and demonstrates it in Fig. S1:

DOI: 10.1038/NMAT3246

We thank the reviewer for bringing this references to our attention and added a discussion using the provided references to support our $B_1$ field simulations (200 ff.).